# mTOR Signalling in Head and Neck Cancer: Heads Up

**DOI:** 10.3390/cells8040333

**Published:** 2019-04-09

**Authors:** Fiona H. Tan, Yuchen Bai, Pierre Saintigny, Charbel Darido

**Affiliations:** 1Division of Cancer Research, Peter MacCallum Cancer Centre, Grattan Street, Melbourne, Victoria 3000, Australia; fiona.tan@petermac.org (F.H.T.); yuchen.bai@petermac.org (Y.B.); 2Univ Lyon, Université Claude Bernard Lyon 1, INSERM 1052, CNRS 5286, Centre Léon Bérard, Centre de recherche en cancérologie de Lyon, 69008 Lyon, France; Pierre.SAINTIGNY@lyon.unicancer.fr; 3Department of Medical Oncology, Centre Léon Bérard, 69008 Lyon, France; 4Sir Peter MacCallum Department of Oncology, The University of Melbourne, Parkville, Victoria 3052, Australia

**Keywords:** mTOR signalling, metabolism, head and neck cancer, mutant genes, biomarkers, targeted therapies, clinical trials

## Abstract

The mammalian target of rapamycin (mTOR) signalling pathway is a central regulator of metabolism in all cells. It senses intracellular and extracellular signals and nutrient levels, and coordinates the metabolic requirements for cell growth, survival, and proliferation. Genetic alterations that deregulate mTOR signalling lead to metabolic reprogramming, resulting in the development of several cancers including those of the head and neck. Gain-of-function mutations in *EGFR*, *PIK3CA*, and *HRAS*, or loss-of-function in p53 and *PTEN* are often associated with mTOR hyperactivation, whereas mutations identified from The Cancer Genome Atlas (TCGA) dataset that potentially lead to aberrant mTOR signalling are found in the *EIF4G1*, *PLD1*, *RAC1*, and *SZT2* genes. In this review, we discuss how these mutant genes could affect mTOR signalling and highlight their impact on metabolic processes, as well as suggest potential targets for therapeutic intervention, primarily in head and neck cancer.

## 1. Background

Head and neck squamous cell carcinoma (HNSCC) is currently the sixth most frequently diagnosed malignancy worldwide [1]. It is the most common cancer of the head and neck, with anatomic subsites spanning the oral cavity, nasopharynx, larynx, oropharynx, and hypopharynx. HNSCC is a heterogeneous disease that harbours complex genetic defects. While the specific multiple risk factors for the development of HNSCC differ depending on the cancer site, chronic tobacco use and alcohol abuse are historically recognised as the main promoting factors associated with the overall occurrence of HNSCC [2,3,4]. Infection with high-risk human papillomaviruses (HPV) has also emerged as a risk factor for a subset of HNSCC (~25% of cases) but has more of a profound role in the development of oropharyngeal cancer [5,6,7]. Nonetheless, HPV positive HNSCCs are shown to have better prognosis compared to HPV negative patients (70–80% versus 25–40%) [8,9,10]. The standard of care for HNSCC patients involves surgery, radiation therapy, chemotherapy and most recently, targeted therapy and immunotherapy. However, these therapies are usually administered in the absence of accurate biomarkers of response, which often leads to treatment resistance, higher systemic toxicities and, in some cases, results in morbidity and mortality. Currently, the only Food and Drug Administration (FDA)-approved targeted therapy for recurrent or metastatic HNSCC patients is cetuximab, a monoclonal antibody that specifically binds and inhibits the epidermal growth factor receptor (EGFR). However, only ~10% of patients demonstrated a beneficial response to cetuximab therapy, while the remainder were at higher risk of relapse [11,12]. Pembrolizumab, an immune checkpoint inhibitor that targets tumour cells expressing high levels of PD-L1 has also been FDA-approved for the treatment of patients with recurrent or metastatic HNSCC. Unfortunately, the rate of pembrolizumab responders is also quite low (~20%) in the absence of patient stratification [13,14,15], and a significant proportion of patients may experience increased tumour growth kinetics (hyperprogressive disease) [16,17]. While significant advances in optimising therapeutic responses have been made, the five-year survival rate has remained between 25 and 60% [18]. Genetic alterations and complex signalling pathways have been shown to drive treatment resistance, allowing for continuous cancer cell survival and proliferation. These mechanisms render most HNSCC patients hard to cure, and therefore there is a need to identify biomarkers of treatment response that will serve to tailor treatment regimens in specific subsets of HNSCC patients. 

## 2. Genomic Alterations in Head and Neck Cancer

The recent application of next generation sequencing to study patient cancer genomes has revolutionized medical oncology. In silico analyses provide great insights into the diverse genomic alterations within each cancer sample, allowing for a functional understanding of the drivers behind deregulated oncogenic pathways and biological mechanisms involved in cancer progression. Importantly, these approaches are being exploited to potentially personalise suitable treatment regimens, tailored towards targeting key oncogenic drivers based on the individual’s mutational profile. On this basis, The Cancer Genome Atlas (TCGA) has been extensively interrogated for a comprehensive genomic characterization of HNSCC, whereby several reports have identified hundreds of mutations in each cancer subtype [19,20,21,22]. This has resulted in common dysregulated pathways being identified across most HNSCC patients. Multiple genetic and epigenetic alterations, including point mutations, deletions, promoter methylation, and oncogene amplification, are strongly triggered by chronic exposure to the major risk factors associated with HNSCC development. Some of the mutated genes frequently associated with HNSCC are *TP53*, *CDKN2A*, *FAT1*, *NOTCH1*, *EGFR, HRAS*, and *PI3KCA* [23,24,25,26,27], and the mutations in these genes are recognised as drivers of tumour development and progression. In HPV^+^ subtype patients, high-risk HPV infection has been associated with the abnormal expression of proteins associated with cell cycle regulation, including p53 and p16 (CDKN2A) [21,28,29]. Functional TP53 inhibits the mammalian target of rapamycin (mTOR) pathway through AMPK in response to cellular stresses and DNA damage [21]. On the other hand, aberrant TP53 allows for persistent mTOR activity and has been associated with poor survival in HNSCC patients [30].

Furthermore, FAT1 is involved in the migration and invasion of HNSCC cells through the activation of the β-catenin pathway [31]. In addition, Notch1 has been reported to play a bimodal role as a tumour promoter and tumour suppressor [22,32,33]. Overall, genetic alterations in the above-mentioned genes and in *EGFR* and *HRAS* often lead to aberrant signalling and deregulation of important proto-oncogenic networks, such as the PI3K–mTOR pathway. For instance, *PIK3CA* mutations that directly activate the PI3K–mTOR signalling pathway have been reported in HNSCC at rates ranging from 2.6% to 19% [21]. Overall, genetic amplifications and overexpression of key proteins responsible for driving mTOR activation underlie the tumour progression that is often observed in cancers, including HNSCC. This review provides a comprehensive analysis of the driver mutations that lead to aberrant mTOR signalling in HNSCC and assesses a number of contemporary inhibitors.

## 3. The mTOR Complex and the Cellular Metabolism

The mammalian target of rapamycin (mTOR) is a serine-threonine kinase that senses growth factor cues, nutrient and oxygen status, and directs appropriate changes to maintain cellular and tissue homeostasis (Figure 1). The mTOR signalling pathway is recognised as a key driver and regulator of cell growth and proliferation, cell survival, metabolism, and protein synthesis. mTOR belongs to the phospho-inositide 3-kinase (PI3K)-related kinase family, and consists of two distinct complexes; mTOR complex 1 (mTORC1) and complex 2 (mTORC2). Whilst both complexes are tightly regulated in a normal context, they are often deregulated in multiple disease-associated metabolic alterations and in cancer development [34,35,36]. In normal conditions, activation of mTOR signalling occurs in response to the binding of specific growth factors to their cognate receptor tyrosine kinases (RTKs), including insulin-like growth factor (IGF), vascular endothelial growth factor (VEGF), platelet-derived growth factor (PDGF), and epidermal growth factor (EGF) [37]. The ligand-activated receptors recruit PI3K, which converts phosphatidylinositol bisphosphate (PIP2) to phosphatidylinositol triphosphate (PIP3), and provides binding sites for phosphoinositide-dependent protein kinase 1 (PDK1). PDK1 then phosphorylates and activates AKT, which in turn phosphorylates several downstream substrates that engage multiple pathways, including mTORC1. On the other hand, mTORC2 has been known to phosphorylate and activate members of the AGC kinase family, including AKT, serum and glucocorticoid-induced kinase (SGK1), and protein kinase C (PKC), whereby inhibition of these kinases results in tumour suppression [38,39,40].

Constitutive mTOR activation is known to promote metabolic changes, including dysregulation in glucose, fatty acid, amino acid, and lipid metabolism. For instance, cancer cells largely rely on glucose as the major source of cellular energy to sustain proliferation and survival. In the context of aberrant mTOR signalling, glucose metabolism is dysregulated as a result of increased synthesis of glucose transporter proteins and glycolytic enzyme activation, followed by lactic acid fermentation even when oxygen is available—a phenomenon known as the “Warburg effect” [41,42]. The Warburg effect links the rewiring of metabolism to sustained cancer cell survival and growth, in which increased glucose uptake and fermentation of glucose to lactate are key processes [43]. In several cancers, including HNSCC, the expression of glucose transporter 1 (GLUT1) is often elevated, and in conjunction with enhanced mTOR signalling (mTORC1 and C2), the pair activates key oncogenic drivers, including c-MYC and HIF-1α [44,45,46,47]. GLUT1 is a protein of the GLUT family, responsible for glucose uptake into the cytoplasm [48]. GLUT1 is negatively regulated by glycogen synthase kinase-3 (GSK-3) that, in turn, exerts its inhibitory effects through a tuberous sclerosis complex (TSC)- and mTOR-dependent pathway [44]. Glycolysis is also upregulated through mTOR signalling via elevated Hexokinase 2 (HK2) expression, which further promotes the activation of c-MYC and HIF-1α [49]. Furthermore, the mTOR signalling stimulates fatty acid synthesis in cancer, via the persistent activation of sterol regulatory element-binding protein-1c (SREBP1c) [50,51]. Although not widely described in HNSCC, Guri et al. observed that elevated lipogenesis correlated with enhanced mTOR activity in hepatocellular carcinoma patients, which in turn facilitated energy production and cancer growth [52]. Overall, deregulated or reprogrammed mTOR signalling is a key signature of cancer cellular metabolism, while the molecular manipulation of the internal and surrounding tumour environment both initiates and sustains cancer cell survival, growth, and proliferation.

## 4. Deregulated mTOR Signalling in HNSCC

The mTOR pathway is known to be hyperactivated in several cancers, including HNSCC, and both mTOR complexes play essential roles in HNSCC tumorigenesis. Interestingly, mTOR deregulation in HNSCC is the most commonly seen genomic alteration (~80–90% HNSCC) involved in aberrant mitogenic signalling, compared to other known pathways such as the JAK/STAT and MAPK, which harbour mutations in less than 10% of lesions [53,54]. 

In vivo analyses of mTOR signalling in HNSCC are commonly studied, and chemically-induced HNSCC mouse models have long been established. These include the widely used carcinogens DMBA-TPA and 4-nitroquinoline-1-oxide (4NQO), which have both been reported to result in persistent mTOR activation, leading to tumour development, and regression is observed after the administration of the mTOR inhibitor rapamycin [55,56,57]. Furthermore, several studies have analysed the effect of rapamycin and mTOR activity in other mouse models, including anal squamous cell carcinoma (SCC) and skin and breast cancers [58,59,60]. Aside from chemically-induced models, genetic mouse models have also been established for mTOR hyper activation. For instance, Sun et al. observed that conditional deletion of Tgfbr1 and Pten in an HNSCC mouse model was associated with the development of sporadic tongue tumours that were driven by mTOR activation [61]. Furthermore, tumour burden was significantly reduced following rapamycin treatment, confirming the role of mTOR in driving HNSCC [61]. Bozec et al. analysed the effect of temsirolimus, a potent mTOR inhibitor, in combination with cetuximab and conventional chemotherapeutic agents (cisplatin and 5-fluorouracil) on orthotopic CAL33 xenografts harbouring *PIK3CA* mutations [62]. The combination therapy was synergistic and resulted in almost complete tumour growth arrest, further associating a profound role between tumorigenesis and mTOR activity [62]. Furthermore, it has been reported that co-targeting mTOR and PD-L1 enhances tumour growth inhibition in a syngeneic oral cancer mouse model [63]. 

mTORC1 interacts with the Rag GTPases, which promote its translocation and activation at the lysosomal surface in response to amino acids [64,65]. Inhibition of mTOR reduced the lysosomal efflux of essential amino acids and converted the lysosome into a cellular depot for them [66]. This process can also be deregulated in cancers where inactivating mutations of the Rag GTPases regulators can lead to hyperactivation of mTORC1, even in the absence of amino acids [67]. mTOR not only has a major role in tumour progression but also plays a role as the central regulator of autophagy. Autophagy is an intracellular process mediated by lysosomes for the breakdown and recycling of damaged cellular components (e.g., organelles, proteins) [68]. In HNSCC, the oral cavity has been known to acquire mutations that are associated with impaired autophagy and correlate with reduced overall survival [69,70]. Whilst a number of inhibitors against the PI3K/AKT/mTOR signalling pathway have undergone extensive preclinical evaluation, the specific mechanism of the action remains elusive and successfully reversing defective autophagy has been variable [71,72]. 

## 5. HPV Status, mTOR Activation and Metabolism in HNSCC 

HPV infection is known to activate mTOR signalling in HNSCC and is further sustained through deregulation of metabolic pathways. For instance, HPV-positive cells utilise mitochondrial respiration, as evidenced by increased oxygen consumption in comparison to HPV-negative HNSCC cells, which exhibit increased glucose metabolism, as evidenced by the over production of lactate [73]. HPV-negative cells express HIF-1α, which is responsible for upregulating downstream mediators involved in glucose metabolism, including hexokinase II (HKII) and carbonic anhydrase IX (CAIX), while HPV-negative cells show greater expression of cytochrome c oxidase (COX) [74,75]. Moreover, as a result of increased lactate and pyruvate production, Jung et al. found that HPV-negative HNSCC cells exhibit advantageous growth, survival and radioresistance [73]. Inhibition of pyruvate dehydrogenase kinase (PDK) sensitises HPV-negative HNSCC to irradiation, which could potentially explain why those tumours are more inclined to have an unfavourable prognosis compared to HPV-positive tumours [9,10]. Therefore, both HPV-positive and HPV-negative HNSCC cells are characterised by deregulated mTOR signalling, which impairs their metabolism and thus sustains the survival and growth of cancer cells in a vicious cycle.

mTOR inhibitors have shown promising anti-cancer effects in HPV-positive HNSCC mouse models. The mTOR inhibitors Rapamycin and RAD001 reduced tumour burden in HPV-positive HNSCC xenografts through the inhibition of mTOR activity [76]. Moreover, HPV E6/E7 mouse models develop SCC lesions with high mTOR activation, and, unsurprisingly, tumour development was abolished using the mTOR inhibitor Rapamycin [56]. Despite the link between HPV infection and mTOR signalling activation with altered metabolic processes, the potential inhibition of both mTOR and HPV-deregulated pathways in HNSCC is still not well explored. 

Furthermore, the E6 and E7 HPV oncoproteins are known to correlate with *PIK3CA* mutations or amplifications in over half of HPV-positive HNSCC, leading to drug resistance. Brand et al. showed that PI3K inhibition resulted in increased expression of the HER3 receptor and, in turn, elevated the abundance of E6 and E7 oncoproteins to promote resistance to PI3K inhibition [77]. This study also assessed the targeting of HER3 with the monoclonal antibody CDX-3379, which resulted in reduced E6 and E7 expression and enhanced the treatment efficacy of PI3K-targeted inhibition. As concluded by the authors, this suggests that co-targeting HER3 and PI3K may be an effective treatment strategy for HPV+ tumours where HER3 and HPV oncoproteins promote resistance to PI3K inhibitors. In addition, Madera et al. inhibited mTOR signalling using metformin, a ubiquitous anti-diabetic drug. This resulted in reduced tumour growth that was driven by the PIK3CA and HPV oncogenes in oral SCC (OSCC) [78]. 

## 6. Validated Mutant Genes Known to Drive Activation of mTOR Signalling in HNSCC 

Mutations in EGFR, *PIK3CA*, and *HRAS*, as well as others found in potential genes such as *EIF4G1*, *RAC1*, *SZT2*, and *PLD1*, can result in aberrant mTOR signalling (Figure 2). Deregulation of mTOR signalling can equally be induced by a loss of tumour suppressors such as *PTEN*, *APC*, and *NF1* [79,80,81]. A hyperactive mTORC1 engages downstream effectors through phosphorylation of the eukaryotic translation initiation factor 4E-binding proteins (4EBP-1), p70 and ribosomal protein S6 kinase (S6K1), promoting tumour development and progression [82,83,84]. In a similar manner, hyperactivation of mTORC2 drives cancer cell survival, proliferation and migration mainly through the oncogenic activation of AKT [34,85,86]. We next discuss the role of these validated and proposed genes (those that have not been well studied in HNSCC) that directly or indirectly activate mTOR signalling in HNSCC.

### 6.1. EGFR-PI3K-AKT-mTOR Pathway 

The hyperactivation of EGFR through various epigenetic and genetic mechanisms is known to activate the PI3K-AKT-mTOR pathway. This is evident in HNSCC patient samples that show high EGFR-mTOR signalling, often associated with poor clinical outcomes [87]. A study conducted by Li et al. demonstrated that a positive feedback loop involving EGFR-mTOR and the inhibitor of nuclear factor kappa-B kinase (IKK)-NF-κB signalling regulates HNSCC cell growth [88]. One regulator of the EGFR-mTOR pathway is the AXL protein, which was shown to dimerize with and phosphorylate EGFR, and to activate phospholipase Cy (PLCy)-protein kinase C (PKC), resulting in the hyper activation of mTOR [89]. Furthermore, genetic alterations of the EGFR gene result in a common cancer-associated variant III (EGFRvIII), in several cancers. Widely studied in gliomas, the EGFRvIII is characterized by the absence of exons 2–7, leading to disruption of the ligand-binding region and is therefore constitutively active in a ligand-independent manner [90]. However, in HNSCC cases the role of EGFRvIII has remained controversial. For instance, although the sample number analysed was low, Sok et al. reported EGFRvIII to be hyperactive in >42% of HNSCC samples (14/33) [91]. However, in a recent study conducted by Khattri et al., it was established that EGFRvIII is rarely seen in HNSCC samples (2/540, 0.37%) and the clinical significance remains unclear [92]. Moreover, EGFR amplification (chromosome 7) has been reported in 11% of HNSCC cases [93,94]. Hashmi et al. observed that EGFR copy number gain occurs in oral leukoplakia and is tightly linked with an increased risk of oral cancer development [94]. 

Aberrant EGFR signalling was also shown to mediate aerobic glycolysis and to upregulate GLUT1 expression. In EGFR mutated lung adenocarcinoma, Makinoshima et al. found that mTOR signalling plays a crucial role in regulating glycolysis and in upregulating GLUT1 localisation [95]. In a panel of EGFR-mutated lung cancer cell lines, mTOR inhibitors significantly suppressed glycolysis and down regulation of GLUT1 by RNAi reduced cell proliferation [95]. Conversely, Chiang et al. found that mTOR signalling contributes to metabolic reprogramming in erlotinib (EGFR inhibitor) resistant lung cancer cells and strongly correlates with poor clinical outcomes of EGFR-mutated lung cancer patients [96]. 

The frequent activation of the EGFR pathway led to the development of EGFR inhibitors targeting receptor function to prevent downstream signalling, including mTOR activation. To date, cetuximab is the only FDA-approved EGFR inhibitor in combination with chemotherapy or with radiation therapy. Tyrosine kinase inhibitors with reversible-binding activity, such as erlotinib and gefitinib, have been disappointing in the head and neck setting, while irreversible-binding Tyrosine Kinase Inhibitors (TKI), including afatinib, appear clinically promising [97,98,99]. The combination of inhibitors targeting both mTOR and EGFR has also emerged as beneficial. For instance, combinatorial treatment targeting mTOR and EGFR has been successful in other cancers, including small cell lung cancers [100,101]. Furthermore, Bozec et al. investigated combined mTOR (temsirolimus) and EGFR (cetuximab) targeting in an orthotopic xenograft model of HNSCC, which culminated in synergistic effects against tumour growth [62]. In agreement, Lattanzio et al. observed a similar result in HNSCC cell lines [100] and Wang et al. also demonstrated reduced tumour burden in both PIK3CA- and RAS-expressing HNSCC xenografts, particularly in cetuximab resistant HNSCC cell lines [53]. Overall, targeting both EGFR and mTOR related pathways could be a promising personalised targeted therapy for HNSCC patients. 

### 6.2. PIK3CA Mutation and PTEN Loss 

Mutations that activate the catalytic unit of phosphoinositide-3-kinase (PI3K) have been implicated in several cancers, including HNSCC. Gain of function mutations of PIK3CA, the most common activator of the PI3K pathway, is detected in approximately 6–20% of HNSCC cases [21,22]. Lui et al. analysed whole-exome sequencing data from 151 tumours and revealed frequent oncogenic mutations in 30.5% (46/151) of the cases affecting the PI3K-mTOR pathway, whereas only 9.3% (14/151) and 8% (46/151) of tumours harboured mutations in the JAK/STAT or the MAPK pathways, respectively [21]. Furthermore, all tumours exhibiting PI3K pathway mutations were advanced (stage IV) cancers, implying a strong role in cancer progression. 

Aberrant PI3K-mTOR signalling was shown to also regulate the properties of key cancer stem cell (CSC) factors, including the sex determining region Y box 2 (SOX2) [102,103]. SOX2 is involved in cancer stem cell (CSC) maintenance and is also associated with increased levels of CSC markers, including aldehyde dehydrogenase (ALDH1) [104]. Keysar et al. characterised the CSC from patient-derived xenografts and defined the molecular features of tumours caused by tobacco smoking and HPV infection [103]. This work unraveled the consequences of deregulated PI3K signalling, such as increased SOX2 translation and expression of ALDH, resulting in enhanced spheroid and tumour formation. This study also observed reduced SOX2 levels after silencing *AKT1* (downstream of PI3K) or *EIF4E* (downstream of mTORC1), suggesting a direct link between SOX2 regulation and PI3K. Additionally, SOX2 knocks down suppressed ALDH transcripts and protein levels. Moreover, Suda et al. revealed that copy-number amplification of PIK3CA, within 3q (found in up to 30% of HNSCC) is associated with a poor prognosis of HNSCC patients [105] and partially overlaps with PIK3CA driving mutations. In addition, it has been shown that PIK3CA mutations are associated with an elevated uptake of glucose and glutamine in colorectal cancer [106], and a similar effect is observed in PIK3CA mutant breast cancer cells [107]. The elevated glucose and glutamine uptakes fuel the growth and progression of tumourigenicity. 

Conversely, inactivation of phosphatase and tensin homologue (PTEN), a potent tumour suppressor and negative regulator of PI3K, also leads to hyperactivation of PI3K-driven mTOR signalling [108]. Although the penetrance of PTEN mutations in HNSCC ranges between 5 and 16%, loss of PTEN expression is observed in 29% of tongue cancers, and loss of heterozygosity of the PTEN locus occurs in 40% of HNSCC tumours [109,110]. Genetic alterations were even lower in SCC of the skin, in which loss of PTEN was mainly due to loss of gene transcription [111,112]. Deletion of the developmental transcription factor Grainyhead-like 3 (Grhl3) induces HNSCC in both humans and mice [111,113,114,115], and GRHL3 functions as a tumour suppressor against SCC of the skin through the direct transcriptional regulation of *Pten* [111,116,117]. Loss of Grhl3 leads to PTEN downregulation and the development of aggressive cutaneous SCC via the activation of the PI3K–mTOR signalling pathway [111]. Inhibition of PI3K/mTOR using BEZ235 was able to prevent the initiation as well as the promotion to malignancy of carcinogen-induced SCC, but was not efficient against the established cancer [118]. Interestingly, mutations in the *PTEN* gene are rare in human skin SCC and common in HNSCC, which could be a prognostic marker for patients with tongue cancer [111,114,119,120]. Moreover, suppression of PTEN in concert with other tumour suppressors, like transforming growth factor beta-receptor 1 (TGFBR1), can also contribute to deregulated PI3K-mTOR signalling. Bian et al. unraveled the relationship between TGF-β signalling and the PI3K-mTOR pathway by conditionally deleting both TGFBR1 and PTEN in HNSCC mouse models using the Cre-LoxP system. Enhanced cell proliferation and decreased apoptosis occurred, which promoted HNSCC tumour development [121]. 

PTEN loss also promotes cancer progression by enhancing glucose metabolism and reducing DNA repair and checkpoint pathways. Martin et al. observed PTEN loss in prostate cancer cell lines and increased pAKT expression and enhanced glucose metabolism, resulting in the survival of tumour cells [122]. Mathur et al. also observed enhanced glutamine metabolism in PTEN mutant breast cancer cells [123]. Conversely, Garcia-Cao et al. showed that transgenic overexpression of PTEN in mice decreased the levels of PFKFB3 (6-phosphofructo-2-kinase/fructose-2,6-biphosphatase 3) and glutaminase, key rate-limiting enzymes responsible for glycolysis and glutaminolysis respectively, and two important metabolic features of tumour cell growth [124]. Together, these data predict that tumours with loss of PTEN function will respond to treatment with inhibitors of glycolysis and glutaminolysis, therefore providing a potential targeted therapy for these tumours.

### 6.3. HRAS 

*HRAS* belongs to the Ras oncogene family, and *HRAS* mutants are known to aberrantly activate mTOR signalling in HNSCC tumours [125]. The Cancer Genome Atlas analysed 279 HNSCC samples and reported that a subgroup of oral cavity tumours had favourable clinical outcomes displaying infrequent copy number alterations in conjunction with activating mutations of HRAS [20]. Nakagaki et al. utilised next-generation sequencing on a cohort of 80 Japanese OSCC patients, and identified *HRAS* mutations in 5% of samples [126]. Su et al. analysed whole-exome sequencing of 120 Taiwanese OSCC patients and identified 11.7% of the samples were positive for *HRAS* mutations [127]. Furthermore, Koumaki et al. identified *HRAS* mutations in 86 OSCC patients (8.6%) of Greek descent, and defined a role for HRAS in driving PI3K-mTOR signalling in OSCC [128]. Adding to the evaluations of several ethnicities, Murugan et al. identified mutant *HRAS* in 10 out of 56 Vietnamese OSCC patients (18%) and associated these events with an advanced tumour stage [129]. In addition to promoting mTOR activation, *HRAS* mutations have been shown to play a role in altering metabolic processes. Zheng et al. suggest that the HRAS transformed breast cancer cell line MCF10a, derived from an early stage cancer, exhibits a profound alteration in glucose metabolism and is strongly regulated by the oncogenic proteins HIF-1α and c-Myc [130]. 

Additional investigations have implicated the HRAS protein in driving resistance to therapies in HNSCC. In a recent study conducted by Ruicci et al., *HRAS* mutant HNSCC cell lines did not respond to PI3K inhibition (BYL719). This inhibitor induced constitutive MAPK signalling suggests feedbacks between MAPK and PI3K, resulting in persistent mTOR activity [131]. Hah et al. also outlined an association between *HRAS* mutations and resistance to the EGFR tyrosine kinase inhibitor erlotinib in a panel of HNSCC cell lines [132]. Likewise, Rampias et al. demonstrated that oncogenic HRAS leads to the activation of MAPK signalling, which results in resistance to cetuximab in HNSCC cells [133]. Furthermore, the authors evaluated a cohort of 55 HNSCC patients, and identified *HRAS* mutations in 7 out of 55 samples (12.7%) that were associated with a poorer response to Cetuximab treatment. 

## 7. Potential Mutant Genes Activating the mTOR Signalling Pathway

### 7.1. EIF4G1

Eukaryotic translation initiation factor (EIF) 4 gamma 1 (EIF4G1) plays a crucial role downstream of mTOR signalling. EIF4G1 functions as a modular scaffold in the translational initiation complex, interacting with EIF3, EIF4A, EIF4E, poly (A)-binding proteins, and MNK1 [134,135]. When mTOR phosphorylates EIF4E-binding proteins (4E-BPs), it releases 4E (EIF4E) to bind to 4G1 and initiate cap-dependent translation [136]. Moreover, the downstream target of mTOR, programmed cell death 4 (PDCD4), disrupts the interaction between 4A and 4G1, leading to translational inhibition [137]. In this study, PDCD4 was shown to be downregulated by miR-21 in HNSCC, suggesting the initiation of mTOR-regulated translation by 4G1. 

Aberrant over-expression of EIF4G1 is tightly linked with the prognosis of several cancers, such as lung squamous cell carcinoma, inflammatory breast cancer, cervical cancers, and nasopharyngeal carcinoma [138,139,140,141]. Although not well studied in HNSCC, the 2015 TCGA analysis of 279 HNSCC patients has provided evidence that EIF4G1 has a high alteration frequency in HNSCC, with 19% of patients harbouring EIF4G1 amplifications and 3.9% somatic mutations [20]. This is supported by studies in breast and nasopharyngeal cancers, indicating that the overexpression of 4G1 facilitates tumourigenesis, malignant transformation, and invasion, while the depletion of 4G1 remarkably leads to the inhibition of cell cycle progression, invasion, and colony formation in vitro and in vivo [139,140,142]. Although there are limited studies testing EIF4G1 inhibitors in HNSCC, small molecule inhibitors have been investigated in other cancer types. For instance, 4EGI1 has been designed to block the interaction of EIF4G1/EIF4E, resulting in decreased translation of oncogenic proteins and abrogating lung tumour growth in vivo [143,144,145,146]. Moreover, SBI-0640756 and RNA aptamers have been designed to directly target 4G1 in melanoma models [147,148]. Overall, despite limited available evidence, inhibitors of EIF4G or EIF4G1/EIF4E interactions could be emerging as novel strategies to indirectly target mTOR signalling in HNSCC. 

### 7.2. RAC1

The Rac family of small GTPase 1 (RAC1) functions downstream of the mTOR signalling to regulate the reorganization of F-actin, lamellipodia formation, and cell motility [149]. mTORC1-mediated activation is essential to increase RAC1 expression, while mTORC2 directly facilitates RAC1 activation via the inhibition of Rho GDP dissociation inhibitor beta ARHGDIB. This results in the initiation of phosphatidylinositol-3,4,5-trisphosphate dependent rac exchange factor 1/2 (PREX1 and PREX2) and T cell lymphoma invasion and metastasis 1 (*TIAM-1*) expression, which all contribute to tumour growth [150,151]. Increasing evidence suggests that RAC1 modulates mTOR activity, whereby the binding of RAC1 to mTOR regulates the plasma membrane localization of the mTORC1/2 complex. This in turn promotes the phosphorylation of mTOR downstream substrates [152,153]. Collectively, RAC1 is therefore considered to be both an upstream and downstream effector of mTOR activity. 

Overexpression of RAC1 is frequently observed in oral, breast, gastric, testicular, and prostate cancers and increased *RAC1* expression is positively associated with cancer progression [154,155,156,157]. Aberrant RAC1 activity was shown to facilitate metastasis of colorectal and lung cancer cells by multiple mechanisms, including epithelial-mesenchymal transition (EMT), migration, and invasion [158,159]. Although not well studied in HNSCC, the TCGA database reports that 3.2% of HNSCC patients harbour RAC1 somatic mutation [20]. Moreover, persistent RAC1 overexpression has been shown to drive resistance to radio/chemotherapy [20,160,161]. In recent years, limited inhibitors targeting RAC1 have been developed. These have included EHop-016 [162] and EHT 1864, which was designed to prevent RAC1-GTP interactions and the RAC1 downstream effectors in order to block RAC1-mediated metastasis [163]. Alongside monotherapy, recent reports suggest that the combined inhibition of RAC1 and mTOR could dramatically increase treatment efficacy against renal cell carcinoma by dephosphorylating the retinoblastoma transcriptional corepressor 1 (RB1) [164]. Although the exact mechanism of the action of RAC1 inhibitors has not been thoroughly explored, their use as single or combination agents seems to have a synergistic effect with mTOR inhibition that could be considered for the treatment of HNSCC. 

### 7.3. SZT2

Deficiency of seizure threshold 2 protein homolog (SZT2) is commonly detected in patients with intellectual disability, epilepsy, and autism [165,166,167,168]. Only recently however, SZT2 was identified as a component of KICSTOR, which negatively regulates the mTOR signalling pathway [168,169]. Interestingly, the *SZT2* gene shows a relatively high somatic mutation frequency (3.6%) in HNSCC in the TCGA database. In addition, low expression of SZT2 is correlated with a low five-year survival rate of HNSCC patients [20]. Future investigation of the SZT2 function is therefore warranted to determine whether it could act as a prognostic factor for HNSCC patients and/or a possible biomarker of response to mTOR inhibitors. 

### 7.4. PLD1

Phospholipase D1 (PLD1) is an established upstream regulator of mTOR signalling [140]. Once activated, PLD1 leads to the accumulation of phosphatidic acid (PA), resulting in mTOR activation via the ERK signaling pathway, an acquired resistance to mTORC1 inhibitors, and a feed-forward loop, resulting in constitutive PLD1 activity [170,171,172]. High *PLD1* expression is frequently detected in various cancers, including glioma, pancreatic ductal adenocarcinoma, colorectal cancer, hepatocellular carcinoma, breast cancer, and melanoma [173,174,175,176,177,178]. Although PLD1 has not been extensively investigated in HNSCC, data from the TCGA show that 20% of HNSCC patients harbour copy number amplification, while 2.9% of patients harbour mutant PLD1. Based on the high percentage of its genetic alteration, we anticipate that PLD1 could function as a driver or a prognostic marker for HNSCC [20].

Multiple inhibitors targeting PLD1 have been developed, such as VU-0155069 and VU-0359595, which directly bind to the N-terminus, allosterically suppressing the catalytic activity of PLD1 [179]. In addition, inhibitors such as Fifi, ML-299, VU-0155056, and VU-0285655-1 show less selectivity by targeting both PLD1 and PLD2 [180]. Kang et al. found that inhibition of PLD1 suppresses the PI3K–mTOR pathway and results in reduced cell proliferation, migration, and invasion in vitro, as well as reduced tumour growth and EMT of patient-derived xenografts in colorectal and hepatocellular carcinoma [175,178]. Since the published literature is establishing a clear relationship between PLD1 and mTOR, further investigations are required to explore the inhibition of PLD1 for mTOR-driven malignancies, as well as the inclusion of PLD1 inhibitors in HNSCC clinical trials. 

## 8. Current Clinical Trials Targeting mTOR in HNSCC

Because multiple mutant genes are directly associated with the oncogenic activation of the mTOR pathway, it is not surprising that multiple clinical trials are currently targeting aberrant mTOR signalling in cancer (Table 1). For instance, the multicentre Phase II trial recruited platinum/cetuximab-refractory HNSCC patients for treatment with the mTOR inhibitor temsirolimus (NCT01172769). Results from this trial indicate that in a total of 40 patients, the treatment was well tolerated, and tumour shrinkage was observed in 13/40 (39.4%) patients [181]. This study indicated that mTOR inhibition alleviates tumour burden, although further molecular analysis is required to identify predictive parameters for temsirolimus guided treatment response. Patients included in this study showed no mutations of KRAS or BRAF. Following from this trial, a Phase II study of temsirolimus in combination with carboplatin and paclitaxel has been conducted on recurrent and/or metastatic HNSCC patients. This resulted in an objective response in 15/36 (41.7%) patients and stable disease progression in 19/36 (52.3%) patients (NCT01016769) [182]. This trial confirmed that a relatively high response can be observed with combination treatment and suggests that genetic alterations associated with aberrant mTOR signalling necessitate further exploration.

Several clinical trials are currently recruiting HNSCC patients for the assessment of other mTOR pathway inhibitors. Following promising results obtained in vitro and in vivo with BYL719, a potent PI3Kα inhibitor, this drug has progressed to Phase II trials for the treatment of recurrent or metastatic HNSCC patients who have previously failed to respond to platinum-based therapy (NCT02145312) [183]. Moreover, in a large multi-centre clinical trial, the Phase II molecular analysis for therapy choice (MATCH) trial tailors personalised inhibitors to each patient’s individual mutational status (NCT02465060). This study includes HNSCC patients with mutations that activate mTOR signalling, who received the inhibitor sapanisertib, which binds to and inhibits both mTOR complexes. Of the targeted therapies related to the mTOR pathway, patients with *PIK3CA* mutations received the PI3K inhibitor taselisib, patients harbouring *EGFR* mutations received the EGFR inhibitor afatinib, while patients with loss or mutated *PTEN* received the PI3K-beta inhibitor GSK2636771. In addition to monotherapies, combination treatments are scheduled with the PI3K/mTOR inhibitor gedatolisib and the cyclin-dependent kinase 4 and 6 (CDK4/6) inhibitor palbociclib, and HNSCC patients are currently being recruited for this Phase I trial (NCT03065062). 

Despite mTOR signalling driving aberrant metabolic processes, there is currently no clinical trial investigating combinational treatment against both mTOR and dysregulated metabolism in HNSCC patients. Nonetheless, there are several studies targeting key transporters involved in metabolism and HNSCC progression. In a recent study conducted by Mehibel et al., the authors investigated the use of simvastatin (which specifically inhibits lipid and cholesterol biosynthesis) and AZD3965 (which inhibits monocarboxylate transporter 1 and results in enhanced glycolysis) in HNSCC cell lines [184]. They found that prophylactic simvastatin lead to the upregulation of xenograft tumour MCT1 expression that effectively primed these cells for MCT1 inhibition using AZD3965. The combination of these inhibitors led to a delay in tumour growth in HNSCC xenograft models and showed no signs of toxicity. Moreover, AZD3965 has been independently assessed in pre-clinical xenograft studies for other cancer types, such as small cell lung cancer, where it was shown to reduce tumour growth via the inhibition of lactate release and glycolysis [185,186]. 

## 9. Conclusions and Perspectives

The mTOR pathway integrates multiple intrinsic genetic alterations and extrinsic cues, leading to aberrant signalling and metabolic alterations. Since the validated and potential mutant genes, as identified from the TCGA dataset, directly affect mTOR activation status in cancer, they could be used as biomarkers for response and mTOR targeted inhibition in a tissue-specific manner. In fact, multiple biomarkers for predicting drug sensitivity have been proposed, including those related to *PTEN* loss, *PTEN* mutations, *NOTCH1* mutation, and EGFR expression in other cancers, but could be further established in HNSCC. Furthermore, the functional characterisation of these mutant genes and the molecular dissection of their associated oncogenic networks could provide targets for combinatorial therapies to alleviate resistance to mTOR inhibition.

## Figures and Tables

**Figure 1 cells-08-00333-f001:**
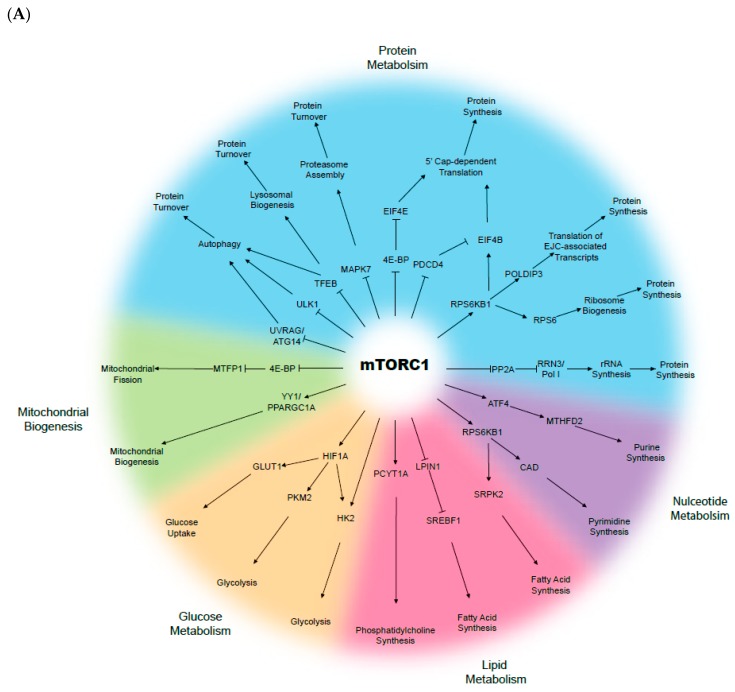
Biological functions of the mammalian target of rapamycin (mTOR): (**A**) mTOR complex 1 (mTORC1) and (**B**) mTOR complex 2 (mTORC2).

**Figure 2 cells-08-00333-f002:**
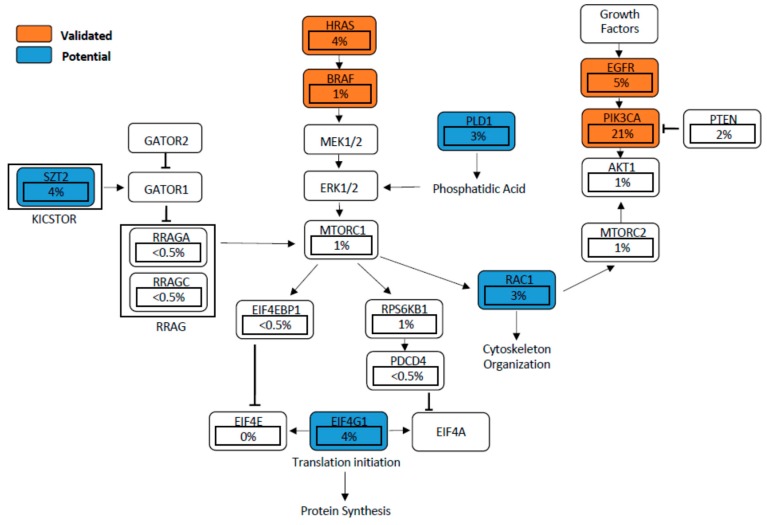
Validated and proposed mutant genes found to deregulate mTOR signaling in head and neck squamous cell carcinoma (HNSCC). Percentages represent frequency of mutations per gene based on The Cancer Genome Atlas (TCGA) dataset (*n* = 297).

**Table 1 cells-08-00333-t001:** Summary of clinical trials targeting mTOR in HNSCC tumours.

Inhibitor	Phase	Status	Targeted Pathway	Targeted Tumour	Reference
Palbociclib + Gedatolisib	I	Recruiting	CDK4/6 + mTOR	Advanced HNSCC	NCT03065062
BYL719	II	Recruiting	PI3K	Recurrent or Metastatic HNSCC	NCT02145312
Everolimus + Palbociclib + Trametinib	I	Recruiting	mTOR + CDK4/6 + MEK	Malignant neoplasms of Oral cavity	NCT03065387
CC-115	I	Active, not recruiting	Dual DNA-PK and TOR kinase	Advanced HNSCC	NCT01353625
PQR 309	I	Active, not recruiting	PI3K/mTOR/AKT	Advanced HNSCC	NCT02483858
Temsirolimus	II	Completed	mTOR	HNSCSC	NCT01172769
Sirolimus	I/II	Completed	mTOR	Advanced HNSCC	NCT01195922
Temsirolimus + Paclitaxel + Carboplatin	I/II	Completed	mTOR	Recurrent or Metastatic HNSCC	NCT01016769

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
