# Peer review of "mTOR Signalling in Head and Neck Cancer: Heads Up"

_cells, 2019, doi:10.3390/cells8040333_

Round 1
Reviewer 1 Report
This review paper glosses over multiple connections to mTOR signaling relating to head and neck cancer with emphasis on current clinical trials that target mTOR in combination with other targets.
I found that the depth of detail on how signaling towards mTOR (and downstream components of mTOR signaling) via the described genes/proteins to be vague. The lack of figures will also make this review hard to interpret for those unfamiliar about mTOR. When the authors mention that a protein activates mTOR, they really need to say how. I also felt the review lacked any type of structure, as I read fragments of information linked to other fragments. I would recommend further editing of the paper to enhance on the structure; again, maybe figures might help with this.
The introduction of mTOR regulation of the Warburg effect on line 104 is an example of a fundamental signaling mechanism of mTORC1 that is not very well described. Here the authors say that Glut1 is elevated with enhanced mTOR signaling, then the pair activate c-MYC and HIF-1alpha. Many questions arise from this. How does mTOR enhance Glut1? What is Glut1? How does Glut1 with mTOR then regulate c-MYC and HIF-1alpha? Was this mTORC1 or mTORC2, or both, as they authors said mTOR? This review is full of sentences of facts, without explanations, which makes this a frustrating review to read in its current state.
My understanding of mTORC1 signaling is that mTORC1 regulates the translation of mTORC1-specific mRNAs. These mRNAs (of which includes C-MYC, CCND1, HIF-1alpha) are inhibited by 4E-BP1 and promoted by S6K1. When mTORC1 is activated, both S6K1 and 4E-BP1 are phosphorylated that leads to enhanced translation of these mRNA. Consequently, when mTORC1 is activated in cancer, then HIF-1alpha protein levels increase that drive transcription of downstream targets such as Glut1, hexokinase, and pyruvate dehydrogenase kinase 1 (PDK1). Higher levels of HIF-1alpha caused higher intake of glucose that is utilized through the pentose phosphate pathway (PPP) via an S6K1 target called CAD to make nucleotide precursors to promote proliferative drive (a hall-mark of cancer). Also, any pyruvate that is made from glucose (glycolysis), is more likely to be converted to lactic acid as the higher levels of PDK1 will block pyruvate dehydrogenase, the enzyme necessary for converting pyruvate to Acetyl-CoA. This is what drives the Warburg effect Therefore, mTORC1 is a driver of metabolic transformation that acts upstream of HIF-1alpha. This paragraph I wrote is quite a different summary of mTOR->HIF-1alpha, if you compare what was said in the review. This is just an example, as there are many more confused and inaccurate sentences within this review.
The authors say that EIF4G1 mutations can lead to aberrant mTOR signalling, which is not entirely true. eIF4G1 is one of many signaling pathways downstream of mTORC1. It is likely that eIF4G1 mutations might lead to enhanced protein translation (a downstream mTORC1 process involved in cell growth control). But I would not say that eIF4G1 leads to aberrant mTOR signalling, as mTORC1 is involved in many processes.
Line 246: How does PI3K increase SOX2 and ADLH? What are SOX2 and ADLH?
Line 250: Why are uptake of glutamine and glucose important for a cancer cell?
Authors should use the correct nomenclature for genes: italics and all in capital letters when referring to a human gene. Proteins are not in italics.
This review needs a least 2 figures that depict the signaling pathways described.
The authors mention that Rac1 is involved in localizing mTOR to the plasma membrane, yet they do not describe mTORC1 localization to lysosomes. I found this odd in a review about mTORC1.
Line 371: how does PA regulate mTORC1?
The conclusions and perspectives was too short in my opinion.
Author Response
We thank the Reviewer for the thoughtful suggestions and comments.
Please refer to the table below for amendments made as recommended by Reviewer 1. Our responses to their questions have been embedded in the revised paper in “Track changes” mode.
Reviewer 1:
Reviewer’s comments | Responses |
1. The introduction of mTOR regulation of the Warburg effect on line 104 is an example of a fundamental signaling mechanism of mTORC1 that is not very well described. | The Warburg effect is now further explained in lines 123–125. |
2. How does mTOR enhance Glut1? What is Glut1? How does Glut1 with mTOR then regulate c-MYC and HIF-1alpha? Was this mTORC1 or mTORC2, or both, as they authors said mTOR? | Responses to questions stated here are provided on lines 128–131. |
3. But I would not say that eIF4G1 leads to aberrant mTOR signalling, as mTORC1 is involved in many processes. | This has been adjusted, please refer to lines 222–223. |
4. Line 246: How does PI3K increase SOX2 and ADLH? What are SOX2 and ADLH? | Please refer to lines 301–303 and 307–309. |
5. Line 250: Why are uptake of glutamine and glucose important for a cancer cell? | Please refer to lines 314–315. |
6. italics and all in capital letters when referring to a human gene. Proteins are not in italics. | This has now been corrected throughout the review. |
7. Needs a least 2 figures that depict the signaling pathways described. | An additional Figure 2 is provided to depict the signalling pathway described. |
8. The authors mention that Rac1 is involved in localizing mTOR to the plasma membrane, yet they do not describe mTORC1 localization to lysosomes. I found this odd in a review about mTORC1. | This has been addressed in lines 168–172. |
9. Line 371: how does PA regulate mTORC1? | This has been addressed in line 450. |
10. The conclusions and perspectives was too short in my opinion. | Please refer to lines 517–523. |

Reviewer 2 Report
The article entitled “mTOR signaling in head and neck cancer: heads up” by
Fiona Tan et al, is a comprehensive review, which summarized the altered mTOR signaling pathway in head and neck cancers. The authors described that the common gain-of-function mutations in EGFR, PIK3CA, and HRAS, or loss-of-function in p53 and PTEN molecules are often associated with mTOR hyperactivation, whereas rare mutations found in the EIF4G1, PLD1, RAC1 18 and SZT2 genes lead to aberrant mTOR signaling. In addition, the authors discussed how these mutant genes affected mTOR signaling, and highlighted their impacts on tumorigenesis. The authors also suggested potential targets of mTOR signaling pathways for therapeutic implications in head and neck cancers. The review article is nicely written, and the reviewer suggests a minor revision for the further improvement of this manuscript.
The specific comments are:
1. In the abstract, the authors stated that “loss-of-function in p53 and PTEN are often associated with mTOR hyperactivation”, however, in the text, there is no specific statement describing how p53 mutations affect mTOR pathways. A recent study by Hui Cheng et al (Cell Reports 25, 1332–1345, 2018), has shown that concurrent 3q26.3 amplification (where PI3KCA, SOX2 and PLD1 located) and TP53 mutation are associated with worse survival of HNSCC patients. Please deliberate more about this point.
2. In “6.2. PIK3CA amplification and PTEN loss”, the authors mainly described the gain function mutations of PI3K. The title should reflect the content.
3. The amplifications and overexpressed of PI3KCA (3q26.33), SOX2 (3q26.33) and PLD1 (3q26.31) are commonly observed in HNSCC and squamous cancers, which genes are very closely located on the chromosome region 3q 26, indicated by recent publications of TCGA and PanCancer TCGA projects (Joshua Campbell et al, 2018, Cell Reports 23, 194–212; Hui Cheng et al., 2018, Cell Reports 25, 1332–1345). Please add the chromosome locations when describe the gene amplification or deletions. Please discuss how theses genetic amplifications and overexpression act together to impact the activation of mTOR signaling pathway.
4. This is a lengthy review, and some paragraphs which are not directly related to mTOR pathway could be removed. For example, “2. Genomic alterations in head and neck cancer” could be shortened and combined with the “6. Common mutant genes known to drive activation of mTOR signaling in HNSCC”.
5. The authors should describe how they define the genes as “rare mutated genes”.
6. Please summarize and discuss how these amplificated and mutated genes or altered phosphorylations of proteins involved in the mTOR pathway could serve or have been developed as the biomarkers for cancer diagnosis, prognosis, or treatment.
Author Response
We thank the Reviewer for the thoughtful suggestions and comments.
Please refer to the table below for amendments made as recommended by Reviewer 2. Our answers to their questions have been embedded in the revised paper in “Track changes” mode.
Reviewer 2:
Reviewer’s comments | Responses |
1. In the abstract, the authors stated that “loss-of-function in p53 and PTEN are often associated with mTOR hyperactivation”, however, in the text, there is no specific statement describing how p53 mutations affect mTOR pathways. A recent study by Hui Cheng et al (Cell Reports 25, 1332–1345, 2018), has shown that concurrent 3q26.3 amplification (where PI3KCA, SOX2 and PLD1 located) and TP53 mutation are associated with worse survival of HNSCC patients. Please deliberate more about this point | This has been addressed in lines 77–79. |
2. In “6.2. PIK3CA amplification and PTEN loss”, the authors mainly described the gain function mutations of PI3K. The title should reflect the content. | The subtitle has been modified to PIK3CA mutation and PTEN loss (line 291). |
3. Please add the chromosome locations when describe the gene amplification or deletions. Please discuss how theses genetic amplifications and overexpression act together to impact the activation of mTOR signaling pathway. | This has been added in lines 86–88. |
4. “2. Genomic alterations in head and neck cancer” could be shortened and combined with the “6. Common mutant genes known to drive activation of mTOR signaling in HNSCC”. | We appreciate the recommendation from Reviewer 2 however, we believe it is best to leave these 2 points separated. |
5. The authors should describe how they define the genes as “rare mutated genes”. | To prevent confusion, this has now been changed to: Common genes à validated genes Rare genes àproposed genes |
6. Please summarize and discuss how these amplificated and mutated genes or altered phosphorylations of proteins involved in the mTOR pathway could serve or have been developed as the biomarkers for cancer diagnosis, prognosis, or treatment. | Please refer to lines 518–523. |

Round 2
Reviewer 1 Report
The review flows better since revision, which will help those readers that are new to the field.
One minor comment: I liked the idea behind the figure depicting the wide-breadth of signaling output from mTOR, however the resolution of the picture and small font size with a rainbow colour background results in a figure that is hard to read. I would recommend having two figures, one for mTORC1 and the other for mTORC2, with bigger font size and a higher resolution image (with no black background on the outside as it is masking the black font on the out-edges of the figure). In the figure Glut1 is a downstream target of HIF1a. Maybe the ? should be replaced by HIF1a.
Author Response
We thank the Reviewer for the minor comment and have made the corrections accordingly.
Figure 1 is now divided into 2 Figures: A) mTORC1 and B) mTORC2. The resolution and font size have been optimised and HIF1A added.